# Effect of Localized Vibration Massage on Popliteal Blood Flow

**DOI:** 10.3390/jcm12052047

**Published:** 2023-03-04

**Authors:** Devin Needs, Jonathan Blotter, Madison Cowan, Gilbert Fellingham, A. Wayne Johnson, Jeffrey Brent Feland

**Affiliations:** 1Mechanical Engineering Department, Brigham Young University, Provo, UT 84602, USA; 2Statistics Department, Brigham Young University, Provo, UT 84602, USA; 3Exercise Science Department, Brigham Young University, Provo, UT 84602, USA

**Keywords:** percussion massage, arterial blood response, Hypervolt, vibration

## Abstract

There is a broad scope of literature investigating whole-body vibration (WBV) effects on blood flow (BF). However, it is unclear how therapeutic localized vibrations alter BF. Low-frequency massage guns are advertised to enhance muscle recovery, which may be through BF changes; however, studies using these devices are lacking. Thus, the purpose of this study was to determine if popliteal artery BF increases from localized vibration to the calf. Twenty-six healthy, recreationally active university students (fourteen males, twelve females, mean age 22.3 years) participated. Each subject received eight therapeutic conditions randomized on different days with ultrasound blood flow measurements. The eight conditions combined either control, 30 Hz, 38 Hz, or 47 Hz for a duration of 5 or 10 min. BF measurements of mean blood velocity, arterial diameter, volume flow, and heart rate were measured. Using a cell means mixed model, we found that both control conditions resulted in decreased BF and that both 38 Hz and 47 Hz resulted in significant increases in volume flow and mean blood velocity, which remained elevated longer than the BF induced by 30 Hz. This study demonstrates localized vibrations at 38 Hz and 47 Hz significantly increase BF without affecting the heart rate and may support muscle recovery.

## 1. Introduction

Over the last 20 years, whole-body vibrations (WBV) utilizing a vibrating platform with lower frequencies in the 20–50 Hz range have become a highly researched topic demonstrating beneficial musculoskeletal or physiological responses [1,2,3,4]. This treatment modality has been used both adjunctively with exercise or in rehabilitation. It has been investigated for its effect on muscle recovery [1,2,3], muscle oxygenation [4,5,6], muscle performance [7,8,9], and wound healing [10,11,12]. Despite the vast amounts of research on WBV application, much less is known about localized vibration effects at similar lower frequencies. In contrast, extensive work has been done on occupational hand–arm vibration syndrome studies at higher frequencies. These studies have shown detrimental effects to finger blood flow, particularly at frequencies greater than 63 Hz [13], which, in turn, have led to conflicting beliefs on the efficacy of localized vibrations as a therapeutic modality. Furthermore, little is known about the primary mechanisms through which vibrations affects the blood flow, but both the frequency and magnitude of vibrations appear to be important contributors to responses in the hand [13].

Whole-body vibration studies have generally shown resultant increases in the blood flow [4], both arterial [14,15,16,17,18] and cutaneous [4,19,20]. The results of these studies vary in the magnitude of response, which may be due to vibration parameters that can differ between studies, such as frequency, amplitude, duration, measurement site and depth (arterial vs. cutaneous), and types of vibration platforms (i.e., vertical vs. oscillating). Uniquely, oscillating vibration platforms using a low frequency (10–30 Hz) have more consistently reported increases in arterial blood flow compared to vertical platforms [4]. The frequency of a vibration has also been shown to influence the magnitude of blood flow responses [4,14,19,21]. For example, Lythgo et al., using standing WBV, demonstrated that the mean blood velocity in the femoral artery increased the most at 30 Hz when comparing 5 Hz increments between 5 and 30 Hz [14]. Meanwhile, Maloney-Hinds et al. reported that 50 Hz increased the skin blood flow more than 30 Hz while uniquely resting the arm on a vertical vibration platform as a form of localized vibration [21]. Thus, the use of vibration improves blood flow; however, the variability of vibration parameters (frequency, amplitude, and duration), along with WBV vs. localized vibrations, leaves an unclear picture as to the optimal combination for arterial vs. cutaneous blood flow responses.

Local vibration treatments have recently gained popularity with the rise of percussion massage devices. These devices allow for the targeted treatment of specific soft tissue as compared to WBV, which affects the entire extremity. A study that surveyed health professionals reported that most respondents believed mechanical percussion would improve the local blood flow (54–69%), followed by the enhancement of pre-exercise excitation (37%) and improvement in the range of motion (31%) [22]. Despite the popularity of these devices, there is a paucity of research utilizing massage guns to support claims that they enhance muscle recovery. Localized vibrations using a vibrating foam roller [23] and another study using a massage gun [24] showed an improved dorsiflexion range of motion in the ankle after treatment of the calf muscles. In other related research, hand and rolling massages offer limited treatment in comparison to local vibrations, as each involve applying pressure and friction to soft tissue. Hand and rolling massage have also been reported to increase the blood velocity [25,26] and skin temperature [27]. However, it is theorized that mechanical pressure can increase blood flow secondary to increasing arteriole pressure [28]. It is believed that local vibration therapy using a massage gun increases the arterial blood flow, but this has yet to be proven. Therefore, the purpose of this study was to examine the effect of frequency and duration of localized percussion massage of arterial blood flow in the lower leg.

## 2. Materials and Methods


Subjects


Twenty-six subjects (fourteen males, twelve females) were recruited from a university population and completed this study (mean age is 22.3 ± 2.3 years). Male subjects had a mean height of 182.6 ± 9.1 cm and average weight of 83.9 ± 18.7 kg. Female subjects had an average height of 168.4 ± 8.4 cm and average weight of 64.9 ± 10.0 kg. Inclusion criteria for the study required that subjects were recreationally active, described for this study as participating in at least 30 min of exercise 3x/week. Additionally, qualified subjects were required to be injury-free in the lower extremities over the past 3 months and without current complaints of lower extremity pain, discomfort, or soreness. Exclusion criteria consisted of a history of cardiovascular issues, peripheral vascular disorders, or taking any type of blood pressure medication. Qualified subjects were asked to avoid participating in a lower body exercise routine within the 4 h prior to each therapy visit and avoid consuming caffeine within 24 h of therapy. All qualified subjects signed an approved institutional IRB consent form (IRB2022-083) in accordance with the Declaration of Helsinki.


Protocol


This was a randomized crossover design. Subjects reported to the lab on 8 different days, with each visit separated by at least 24 h. Each subject underwent all therapeutic conditions and control conditions consisting of three frequencies (30 Hz, 38 Hz, and 47 Hz) and a control condition (C) for both a 5-min and 10-min duration in randomized order. Subjects began each session by having a 3-lead ECG connected to them to measure their heart rate. The ECG was directly linked with the GE Logiq S8 ultrasound unit (GE Healthcare, Chicago, IL, USA). Subjects lay prone on a treatment table for 10 min with their ankles placed on a small foam roller to allow the feet to hang relaxed and the knee to be mildly bent to allow the calf muscles to be relaxed without additional stretch caused by laying with the knee extended (see Figure 1).

Baseline measurements of blood flow were then taken, followed by vibration therapy to the calf area using a Hypervolt percussion massage gun (Hyperice, Irvine, CA, USA) set to a predetermined frequency and time duration. The massage gun was swept smoothly at an even tempo of approximately two seconds (proximal–distal–proximal) over the gastrocnemius and soleus muscles, with no added pressure other than the weight of the device. For the control condition, the massage gun was off (no percussive vibration), but the head of the device was moved over the muscle belly similar to the vibration conditions. Post-therapy blood flow measurements were taken at 13 different time points: immediately after vibration was completed, then once per minute for 5 min, and, finally, another 7 measurements taken every 2 min until 19 min post-vibration.


Measurements


A GE Logiq S8 with pulse wave doppler ultrasound and a 9L transducer in vascular mode were used to measure the arterial blood flow in the popliteal artery below the knee. This is consistent with previous methods used for measuring arterial blood flow [14,16] and has been shown to be reliable [29]. The outcome measures included heart rate (bpm), mean blood velocity (cm/s), arterial diameter (cm), and volume flow (mL/min). Five cardiac cycles were used to determine the mean velocity and volume flow. All measures other than heart rate were normalized to the baseline value for that specific condition. This allowed for the comparison of conditions within the same subject. All participants completed the study within four weeks of starting.

Hypervolt states the frequencies of the gun are 30 Hz at level 1, 40 Hz at level 2, and 53 Hz at level 3. Using an omega HHT41 digital stroboscope (Omega, Stamford, CT, USA), we determined the actual frequencies of the Hypervolt gun were 30 Hz, 38 Hz, and 47 Hz. We also verified that the Hypervolt massage gun head has a measured amplitude of 2.56 cm and weight of 1.14 kg.


Statistical Analysis


An initial statistical analysis was performed for the arterial diameter, volume flow (VF), and mean velocity (MV). This analysis revealed insignificant changes in the arterial diameter. Similarly, the heart rate measurements showed little change; thus, only MV and VF were used for further statistical analysis. The data for all 26 subjects was averaged at each time point to analyze the average effect of each condition. Using a cell means mixed model to appropriately account for both within-subject and between-subject variability, contrasts were created for a more detailed analysis. The condition effect was analyzed by comparing the pre- and post-vibration values for VF and MV, and the linear and quadratic recovery effects were analyzed.

The effect of each condition requires a single test. Therefore, to analyze 8 conditions’ pre- to post-values for both VF and MV requires 16 tests. Multiple tests artificially increase the significance level (α) of the *t*-tests. Using a significance level (α) of 0.025, the original *t*-value was ±2.064. To compensate for the α inflation due to 16 tests, Bonferroni correction was used, resulting in a new *t*-value of ±2.99. Similarly, for the 4 conditions with significant condition effects (38 and 47 Hz at both 5 and 10 min), the linear and quadratic recovery effects were tested for both MV and VF, resulting in another 16-test analysis. The linear and quadratic effects for the 10-min control condition were not analyzed to avoid further alpha inflation and because there was no increase in the blood flow to analyze for return to the baseline. Thus, a *t*-value with an absolute value greater than 2.99 was used to determine the significance for all the results of the *t*-tests and to maintain an alpha level of 0.05.

## 3. Results

Table 1 shows the *t*-values of the pre- and post-effects of the vibration, the linear effect, and quadratic effect of recovery for each variable and each condition.

These results show that vibrations have a consistent effect pre- and post-vibration for MV and VF. It also clearly reveals that the control and 30 Hz 5-min conditions do not have a statistically significant effect on MV or VF. Interestingly, the control for 10 min only had a significant negative effect for MV, while 30 Hz for 10 min neared significance for both MV and VF. Finally, the *t*-tests showed that the conditions at 38 Hz and 47 Hz result in a significant increase of MV and VF for both the 5-min and 10-min conditions. Figure 2 and Figure 3 show the MV and VF responses, respectively, for all 8 conditions. Post-vibrations in these figures are represented as time 0 min.

Figure 2 and Figure 3 show that MV and VF continue to increase for up to 3 min post-vibration. They also depict the trend that each variable gradually recovers toward its baseline over the remaining measured time periods. The *t*-tests revealed that there is a significant linear effect for all conditions of both VF and MV. Figure 4 and Figure 5 show the VF data with a linear best fit for the post-vibration data. Finally, the *t*-tests show that the quadratic recovery effect is not significant for any condition.

## 4. Discussion

The objective of this work was to determine if and how much local vibrations, using a massage gun, increase arterial blood flow. The pre-post-analysis of VF and MV show a significant increase that is dependent on frequency and time, with greater blood flow at a higher frequency and longer durations. The VF increase for 38 Hz at 5 min and 10 min was 24% and 32%, respectively. The 47 Hz conditions saw VF increases of 31% for 5 min and 47% for 10 min of vibrations.

The data clearly show that VF and MV continue to increase for several minutes post-vibration using both 38 Hz and 47 Hz. For the 47 Hz 10-min condition, both VF and MV peaked at 3 min post-vibration, while the 47 Hz 5-min and 38 Hz 10-min conditions peaked at 2 min. Finally, the 38 Hz 5-min condition peaked at 1 min post-vibration. It is reasonable to assume that, with a larger sample size, this delayed effect would be more pronounced. While we only measured post-vibration blood flow for 19 min, Figure 4 and Figure 5 show that the high-frequency and duration conditions result in an elevated blood flow that demonstrates a longer return to baseline. This residual effect is important to consider for either recovery treatments after an activity or the application of pre-activity use in athletes who want their blood flow elevated in a particular muscle group prior to competition.

Further observation of the linear return to baseline of each condition provides some interesting insights into the effects of frequency and duration. The effect of duration has a smaller impact than frequency on the slope of the blood flow return to baseline. This is particularly evident when comparing 30 Hz to 38 Hz as compared to 38 Hz vs. 47 Hz. The slope for the linear recovery of VF with 47 Hz for 10 min is −1.7%/min and 47 Hz for 5 min is −1.6%/min. For VF with 38 Hz, the slopes were −1.6%/min and −1.3%/min for 10 min and 5 min, respectively. Thus, the rate of VF recovery for these two frequencies is similar. Finally, for VF with 30 Hz, the slope for 10 min was −1.1%/min and −0.9%/min for 5 min. Therefore, within each frequency, the VF slopes are comparable between the 5-min and 10-min conditions. This also indicates that, as the frequency increases from 30 Hz to 38 Hz, so does the steepness of the slope, suggesting that, as the blood flow becomes more elevated, the blunting or recovery of the blood flow to baseline may occur at a faster rate. Future studies should carry out post-vibration blood flow measures longer to determine how the residual blood flow differs by frequency and time. Variations in the rate of return to the baseline could affect the practical application of localized vibration.

These findings of increased blood flow at frequencies above 30 Hz are supported by Maloney-Hinds et al. [21], who investigated skin blood flow responses to the forearm. In their study, participants had their arm passively vibrated by resting the forearm on a WBV platform at 30 Hz and 50 Hz for 10 min. The blood flow was measured using a laser Doppler flow meter between each minute of vibration for 10 min and for 15 min post-vibration. Their study found that 30 Hz and 50 Hz vibrations both resulted in significant increases in the blood flow compared to baseline. Although not significant, their data showed responses at 50 Hz were continually higher than 30 Hz.

Our data show 30 Hz did not significantly affect VF or MV pre-post-vibration, although 5 and 10 min demonstrated an average pre-post-increase in VF of 0% and 18%, respectively. This increase at 30 Hz for 10 min is just below the significant *t*-value, and we postulate that, with a larger sample size, this may reach significance. Furthermore, the peak increase in VF for 30 Hz was 10% and 38% for 5- and 10-min conditions, respectively. This further supports our finding that longer condition durations, up to 10 min, increases VF. Future studies should observe the effect of even longer durations to determine if there is a duration ceiling.

The control condition resulted in decreased VF, as subjects were at rest with very little stimulus, and this became significant at 10 min. This is likely because blood flow decreases in a muscle at rest. The weight of the massage gun exerted a pressure of 0.79 psi while using a flat, round massage head. This likely resulted in minimal tissue deformation and possibly did not exert enough pressure to induce an increase in blood flow responses.

Heart rate measurements were collected to verify that any changes in blood flow were due to local vibration stimulation and not excitation of the cardiovascular system. The heart rate was averaged for each subject across the eight conditions. Only 4 out of 26 subjects demonstrated minimal increases of between 1 and 3 bpm, which is within the normal fluctuation range, while 22 subjects maintained or slightly decreased their average heart rate. These data indicate that vibrations of the calf muscles did not increase the heart rate. This is consistent with other studies that applied WBV and measured the blood flow and heart rate [16,18]. Similarly, the popliteal diameter was measured to determine if the vibrations caused arterial dilation. The diameter measurements were also averaged over the eight conditions for each subject, and only five subjects saw changes in the diameter over 2.5% from pre- to post-vibration. These results were insignificant in our initial statistical analysis. This finding is consistent with a study by Kerschan-Schindl et al., where the systolic area of the popliteal artery reportedly was not significantly changed after WBV exercise [18]. Therefore, changes in the heart rate and popliteal diameter do not appear to have affected our results.

The exact mechanisms of how vibrations cause increased blood flow are still unclear. The results presented here clearly indicate that local vibrations increase arterial blood flow. It is also evident that local vibrations result in a local increase of flow without raising the heart rate, meaning only local blood flow is being excited. Furthermore, the popliteal artery diameter is not significantly affected by vibrations, which implies the physiological response is more locally induced with mechanical stress. It is possible that the mechanical pressure of the therapeutic condition causes an increase in arterial pressure, which, when released, may increase the blood flow [26]. Furthermore, vibration stimulus may result in muscle contraction [30], which, in turn, causes vasoconstriction and increasing arterial back pressure [31]. It is understood that exercise results in the release of vasodilators and an increase in arterial blood flow [32]. Additionally, it has been shown that exercise with vasodilator blockage results in even higher arterial blood flow [32]. Future works could block hormone receptors with antihistamines and measure the blood flow response to determine if the hormone response is primarily responsible for vibrations increasing the blood flow.

The results of this study are limited to healthy university-aged and recreationally active young adults. Any extrapolation of these results to a particular pathology or clinical population such as those with ischemia, diabetes, or musculotendinous injuries requires future investigations to establish efficacy in these populations. Thus, this intervention should not be used as treatment for medical pathological populations until further clinical evidence is provided. While little evidence exists for manual massage-related effects on blood flow and recovery, if increases in the blood flow were a significant factor, a recommendation for light exercise would produce even greater increases in blood flow. However, our results suggest that localized vibration therapy in healthy young active adults can potentially increase blood flow to a particular region or muscle group without the metabolic demands of normal exercise. Those using such devices should note our results are further limited to 5- and 10-min applications, and there is no evidence yet to suggest that increasing the duration will enhance the blood flow effects. For future studies, it is also noteworthy to mention that our study found anecdotal evidence that higher frequencies were often reported to be uncomfortable.

## 5. Conclusions

This study demonstrates that, for healthy university-aged adults, localized percussion vibration using a massage gun increases VF and MV and that higher frequencies and longer durations of vibrations lead to greater increases in both. The data also revealed there is a time delay between the end of the vibration condition and the peak increase in VF and MV. This delay becomes longer with increasing the frequency and duration. Furthermore, we found a strong linear return to the baseline effect with similar VF rates for 38 Hz and 47 Hz and a lower rate for 30 Hz. We also concluded, increases seen in popliteal artery blood flow were not influenced by the heart rate and popliteal diameter. Future research is needed to determine the relationship between frequency and duration effects on blood flow when using massage guns. Research on mechanical or local hormone responses to localized vibrations (such as histamine) should be further investigated to better determine the primary mechanisms contributing to blood flow responses after localized vibrations using a massage gun.

## Figures and Tables

**Figure 1 jcm-12-02047-f001:**
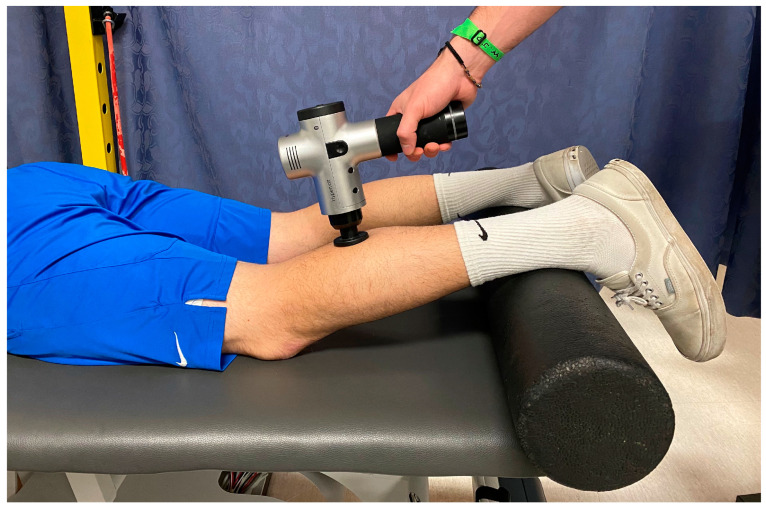
A representative subject is shown to demonstrate the position of each subject when therapy was administered.

**Figure 2 jcm-12-02047-f002:**
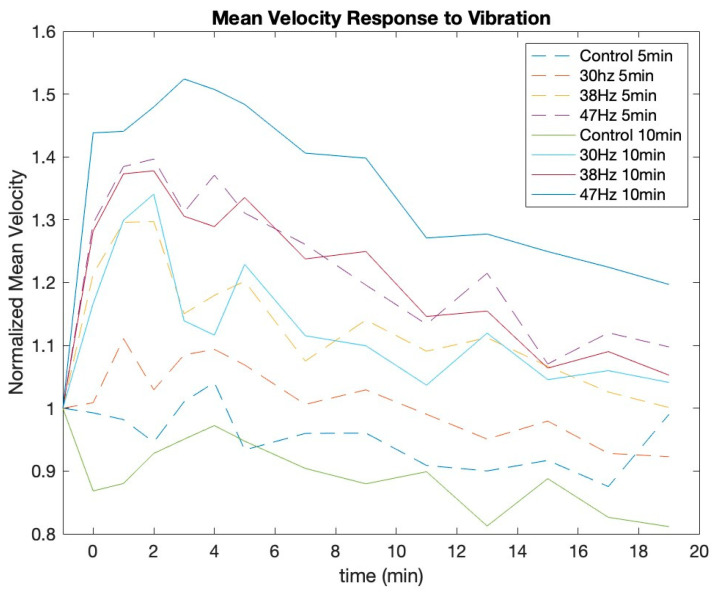
Normalized mean velocity response to each vibration condition.

**Figure 3 jcm-12-02047-f003:**
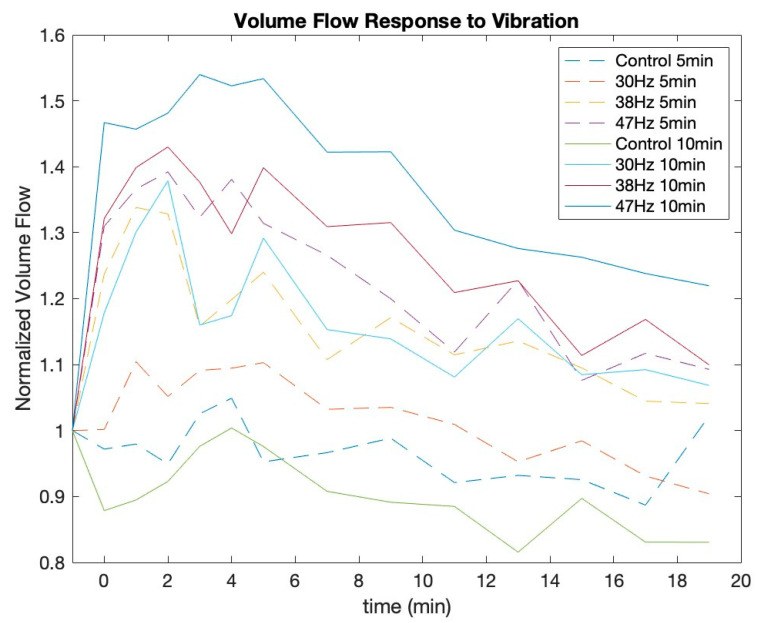
Normalized volume flow response to each vibration condition.

**Figure 4 jcm-12-02047-f004:**
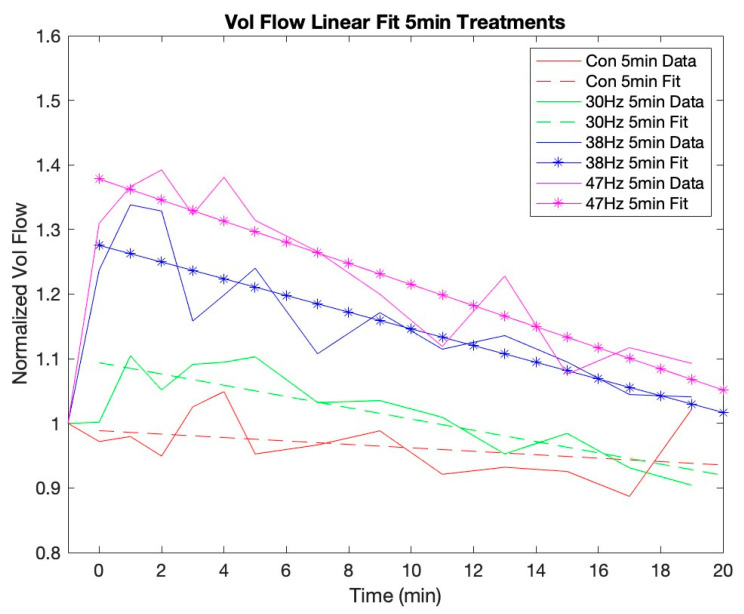
Linear recovery for volume flow with 5-min conditions. Note: 38 Hz and 47 Hz 5 min fit denotes statistical significance of linear data fit for post-vibrations.

**Figure 5 jcm-12-02047-f005:**
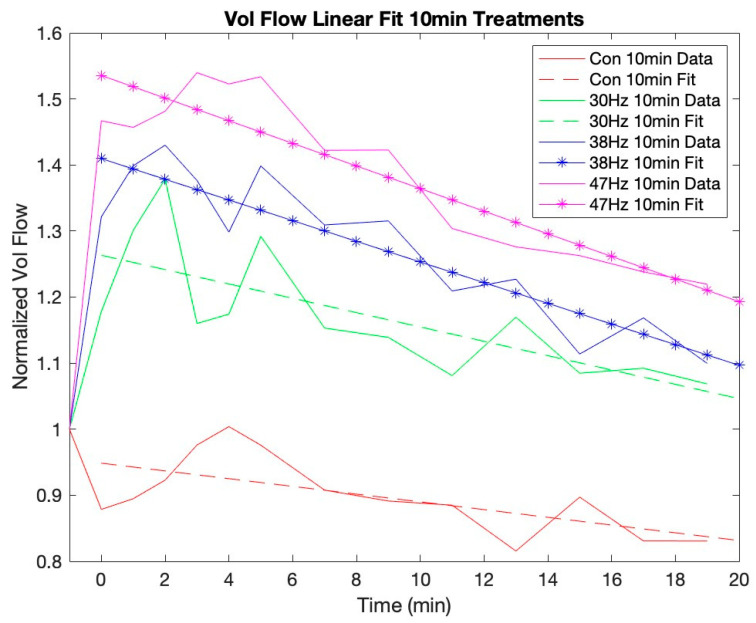
Linear recovery for volume flow with 10-min conditions. Note: 38 Hz and 47 Hz 10 min fit have statistical significance of linear data fit for post-vibrations.

**Table 1 jcm-12-02047-t001:** *T*-values for pre- and post-vibration effect, linear effect, and quadratic effect of recovery for both mean velocity and volume flow of each condition. The corrected *t*-value for significance is greater than |±2.99|.

	Mean Velocity *t*-Values			Volume Flow *t*-Values		
Therapeutic Condition	Pre-Post	Linear	Quadratic	Pre-Post	Linear	Quadratic
Control 5 min	−0.194			−0.622		
30 Hz 5 min	0.227			0.008		
38 Hz 5 min	4.145 *	−7.878 *	0.749	4.287 *	−7.299 *	0.821
47 Hz 5 min	4.766 *	−8.112 *	0.328	4.696 *	−7.700 *	0.211
Control 10 min	−3.167 *			−2.721		
30 Hz 10 min	2.858			2.753		
38 Hz 10 min	4.755 *	−8.897 *	−0.609	4.816 *	−7.296 *	−1.074
47 Hz 10 min	7.016 *	−8.444 *	−1.254	6.996 *	−7.986 *	−1.258

Note: * indicates statistically significant result.

## Data Availability

The data presented in this study are available in the Appendix A of this work.

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
