# Peer review of "Effect of Localized Vibration Massage on Popliteal Blood Flow"

_jcm, 2023, doi:10.3390/jcm12052047_

Round 1

Reviewer 1 Report

Inclusion and exclusion criteria are indicated in the text not prominently.

It will be good and very informative if there is a paragraph for the conclusion.

Registration numbers of the devices used must be Indicates in the text.

All subjects included in this study were healthy and the use of the term "treatment" is incorrect. I think it would be more correct to use other terms, for example, applying a vibrational effect, etc.

In statistical analysis: Thus, a t-value of ±2.99 was used to determine significance for all the results. It is not clear. It will be better if you clarify information. For example;  t>2,99, t<-2,99 ?.

In lines 264-267: “The results of this work have important clinical implications. Populations that could benefit from treatments to increase blood flow include those with ischemia, diabetes, and atherosclerosis, although further research would be needed to show if similar BF responses would occur in subjects with compromised vascularity”.

The results obtained cannot be attributed to the treatment of patients, because research was conduct only in young recreationally active university students in mean age 22.3 ± 2.3 years.

It can be indicated that further studies of the effectiveness and safety of this method in the treatment of patients with various diseases are recommended.

It should be pointed out that this method of intervention should not be used in the treatment of patients until sufficient clinical trials have been carried out.

Author Response

Thank you for your revisions, especially the emphasis on not extending the results of our study to populations that need treatment. We are grateful for your thorough review of our manuscript.

1.Inclusion and exclusion criteria are indicated in the text not prominently.

Lines 83-89 now specifically indicate the inclusion/exclusion criteria for our study

2. It will be good and very informative if there is a paragraph for the conclusion.

Lines 342-355 now comprise the conclusion section of the manuscript

3.Registration numbers of the devices used must be Indicates in the text.

We are unsure what was meant by registration number of the devices used. We haven’t seen registration numbers required by typical journals nor in the author guidelines for this journal. We have included the make and model of the devices used and included the company information for each device. This should be sufficient to find any further information required on the devices used in this study.

4. All subjects included in this study were healthy and the use of the term "treatment" is incorrect. I think it would be more correct to use other terms, for example, applying a vibrational effect, etc.

We have modified the term “treatment” when referring to our study and changed it to either “therapy”, “therapeutic condition”, or “condition”.

5.In statistical analysis: Thus, a t-value of ±2.99 was used to determine significance for all the results. It is not clear. It will be better if you clarify information. For example;  t>2,99, t<-2,99 ?.

Line 165 the description of the significant t-value has been changed to say “a t-value with absolute value greater than 2.99” to clarify. Also the caption of table 1 was similarly modified.

6. In lines 264-267: “The results of this work have important clinical implications. Populations that could benefit from treatments to increase blood flow include those with ischemia, diabetes, and atherosclerosis, although further research would be needed to show if similar BF responses would occur in subjects with compromised vascularity” The results obtained cannot be attributed to the treatment of patients, because research was conduct only in young recreationally active university students in mean age 22.3 ± 2.3 years. It can be indicated that further studies of the effectiveness and safety of this method in the treatment of patients with various diseases are recommended. It should be pointed out that this method of intervention should not be used in the treatment of patients until sufficient clinical trials have been carried out.

Lines 327-332 address that the results of this study should not be extrapolated beyond young, healthy adults and that this intervention should not be used on populations requiring treatment until further clinical evidence is provided.

Reviewer 2 Report

This is a well designed and controlled study which provides evidence for modest changes in lower leg arterial blood flow following localized vibratory massage at various frequencies. 

Given the nature of massage related claims for efficacy it would be prudent for the article to specifically include caveats as the the potential benefits of such massage, particularly for relatively healthy populations and athletes. Specifically, there is currently (including the present study) little evidence for increased blood flow related effects of massage/vibration of muscle healing, post-exercise recovery or performance in healthy populations. Indeed, if increases in muscle blood flow were a significant factor, then recommendations for light exercise or muscle contractions (which would elicit significantly more muscle blood flow than vibration would be therefore far more effective) should be prescribed rather than vibration as measured in this study. This type of commentary needs to be included in the manuscript to ensure that the results of the study are not over interpreted and the efficacy of this type of treatment is not "over-sold". Please include a paragraph to this effect on the "limitations" of these findings.

Author Response

Thank you for your thorough revisions of our manuscript and particularly the comments on providing more detailed limitations of our results.

Lines 327-341 comprise a paragraph on the limitations of this study. This paragraph specifically addresses that the effects of increased blood flow have little evidence and that for healthy populations light exercise is a more effective way of increasing blood flow compared to local vibration. Also we addressed that the results of this study are limited to healthy, active, young adults and further clinical evidence is needed to find the effect of this intervention on other populations.